# The Role of Checkpoint Inhibitor Expression Directly on Exfoliated Cells from Bladder Cancer: A Narrative Review

**DOI:** 10.3390/diagnostics13193119

**Published:** 2023-10-03

**Authors:** Luca Di Gianfrancesco, Alessandro Crestani, Antonio Amodeo, Paolo Corsi, Davide De Marchi, Eugenio Miglioranza, Giuliana Lista, Francesca Simonetti, Gian Maria Busetto, Martina Maggi, Francesco Pierconti, Maurizio Martini, Isabella Monia Montagner, Debora Tormen, Antonio Scapinello, Filippo Marino, Angelo Porreca

**Affiliations:** 1Department of Urology, Veneto Institute of Oncology (IOV)—IRCCS, Headquarter of Castelfranco Veneto, 35128 Padua, Italy; alessandro.crestani@iov.veneto.it (A.C.); antonio.amodeo@iov.veneto.it (A.A.); paolo.corsi@iov.veneto.it (P.C.); davide.demarchi@iov.veneto.it (D.D.M.); eugenio.miglioranza@iov.veneto.it (E.M.); giuliana.lista@iov.veneto.it (G.L.); francesca.simonetti@iov.veneto.it (F.S.); angelo.porreca@iov.veneto.it (A.P.); 2Department of Urology, University of Foggia, 71122 Foggia, Italy; gianmaria.busetto@unifg.it; 3Department of Urology, Sapienza University, 00185 Rome, Italy; martina.maggi@uniroma1.it; 4Department of Pathology, Fondazione Policlinico Universitario Agostino Gemelli IRCCS, University of Sacred Heart, 00168 Rome, Italy; francesco.pierconti@unicatt.it; 5Department of Pathology, University of Messina, 98122 Messina, Italy; maurizio.martini@unime.it; 6Anatomy and Pathological Histology Unit, Veneto Institute of Oncology IOV—IRCCS, 35128 Padua, Italy; isabellamonia.montagner@iov.veneto.it (I.M.M.); antonio.scapinello@iov.veneto.it (A.S.); 7Department of Urology, Fondazione Policlinico Universitario Agostino Gemelli IRCCS, University of Sacred Heart, 00168 Rome, Italy; dr.filippomarino@gmail.com

**Keywords:** bladder cancer, PD-L1, urinary biomarker, diagnostic tool, prognostic tool

## Abstract

Bladder cancer (BCa) is a common type of cancer that affects the urinary bladder. The early detection and management of BCa is critical for successful treatment and patient outcomes. In recent years, researchers have been exploring the use of biomarkers as a non-invasive and effective tool for the detection and monitoring of BCa. One such biomarker is programmed death-ligand 1 (PD-L1), which is expressed on the surface of cancer cells and plays a crucial role in the evasion of the immune system. Studies have shown that the PD-L1 expression is higher in BCa tumors than in healthy bladder tissue. Additionally, PD-L1 expression might even be detected in urine samples in BCa patients, in addition to the examination of a histological sample. The technique is being standardized and optimized. We reported how BCa patients had higher urinary PD-L1 levels than controls by considering BCa tumors expressing PD-L1 in the tissue specimen. The expression of PD-L1 in urinary BCa cells might represent both a diagnostic and a prognostic tool, with the perspective that the PD-L1 expression of exfoliate urinary cells might reveal and anticipate eventual BCa recurrence or progression. Further prospective and longitudinal studies are needed to assess the expression of PD-L1 as a biomarker for the monitoring of BCa patients. The use of PD-L1 as a biomarker for the detection and monitoring of BCa has the potential to significantly improve patient outcomes by allowing for earlier detection and more effective management of the disease.

## 1. Introduction

The bladder cancer (BCa) represents a model to analyze predictors and response mechanisms of the immune system.

Historically, urinary cytology is the most used non-invasive diagnostic tool for BCa and it remains a fundamental exam (as an adjunct to endoscopy and radiology) both in diagnosis and in follow-up in high-grade tumors.

One of the limits of urinary cytology is represented by the user-dependent interpretation [1,2]. Moreover, the final report might be negatively affected by ongoing urinary tract infections, low cellularity, the presence of stones, or previous intravesical treatments; however, in experienced hands the specificity is more than 90% [1]. Thanks to urinary cytology, abnormal cells in urine are detected and a diagnosis of urinary tract cancer is performed. A positive result can indicate a urothelial carcinoma anywhere in the urinary tract; however, a negative results does not exclude BCa [3]. By using voided urine or bladder-washing specimens for exfoliated cell analysis, the sensitivity for the detection of high-grade tumors is up to 84%, and up to 100% in the case of carcinoma in shut (CiS), but very low in case of low-grade tumors (16%) [4,5].

Moreover, one of the perspectives of urinary cytology is represented by the opportunity to analyze common aberrations of chromosomes in urothelial tumors [6]: among these we surely might prospectively consider and investigate the expression of checkpoint inhibitors on exfoliated urinary cells.

The analysis of the expression of programmed death-ligand 1 (PD-L1) has so far focused only on the histological specimen, which requires the use of an invasive method (urethrocystoscopy and/or the endoscopic resection of bladder neoplasia) with considerable discomfort for the patient [7]. To date, the role of checkpoint inhibitors on exfoliated cells has not been directly evaluated and the evidence is scant regarding this issue, which is different for other cancers. The developments in PD-L1 analysis on exfoliate urinary cells might lead a better understanding of the BCa molecular aspects, contribute prognostic information about BCa’s natural history, and affect and lead the subsequent clinical management of the disease, such as the development of targeted intravesical therapies with checkpoint-inhibitor-based drugs. The proposal and innovation is to introduce the analysis of PD-L1 expression directly on exfoliated cells in terms of non-invasive diagnostic and prognostic tools for non-muscle-invasive BCa.

## 2. Advantages of Urinary Cytology

Different cytology tests from different organs allow us to achieve sufficient diagnostic information which is useful in guiding treatments.

The cytology of urine sample is an ordinary, cheap, widely used, and reproducible test with a well-established analysis standard [8].

Cytology represents both a screening (such as in cervical cancer) and a follow-up (including BCa) tool for several malignancies, and sometimes even a prognostic tool in diagnosis (such as in breast cancer).

The four main characteristics of urinary cytology analysis are safety, ease, quickness, and cost-effectiveness [9].

## 3. Significance of PD-L1

PD-L1 is known as B7 homolog 1 (B7-H1) or CD274 and represents a transmembrane protein with the role of downregulating the immune response by binding to its two receptors PD-1 and B7-1 (CD80) [10].

PD-1 is an inhibitory receptor which is expressed on T-cells after T-cell activation in case of chronic stimulation (such as cancer or chronic infection) [11]. When PD-L1 binds to PD-1, there is the inhibition of the proliferation of T-cells, the production of cytokine, and cytolytic activity, with the subsequent exhaustion or inactivation of the function of T-cells. When the PD-L1 binds to CD80 on T-cells and antigen-presenting cells (APCs), there is the downregulation of immune responses, with inhibition in both T-cell activation and cytokine production [12] (Figure 1).

The PD-L1 expression was analyzed in both immune cells and tumor cells [13,14]. The aberrant expression of PD-L1 on tumor cells was showed to inhibit anti-tumor immunity, with subsequent immune evasion. Hence, the interruption of the PD-L1/PD-1 pathway is a promising strategy to reinvigorate tumor-specific T-cell immunity suppressed by PD-L1 expression in the microenvironment of the tumor.

Several cancers including lung, melanoma, urothelial, ovarian, and colorectal express PD-L1, with a prevalence from 12% up to 100% according to tumor type, anti-PD-L1 clone, and cut-off ranges for positivity [15].

## 4. PD-L1 Expression and Cytological Analysis in Other Experiences

Hendry et al. evaluated how adequate tumor cellularity is essential for accurate PD-L1 immunohistochemistry assessment on cytology cell-block specimens. The authors focused on the PD-L1 immunohistochemistry (IHC) as a biomarker with a predictive role for patients with non-small cell lung cancer (NSCLC), and its use for treatment based on anti-PD-1 immune checkpoint inhibitor therapy. The authors investigated the concordance between NSCLC PD-L1 IHC assessed on histology and cytology specimens and the impact of tumor cellularity. The authors performed the PD-L1 IHC concurrently on both specimens by using the SP263 assay kit on the Ventana Benchmark Ultra staining platform and it was scored by two experienced pathologists. The authors reported good overall agreement between matched cytology and histology specimens (intraclass correlation coefficient = 0.653, *n* = 58), with a markedly increased agreement the in case of analysis limited to cell blocks with >100 tumor cells (intraclass correlation coefficient = 0.957, *n* = 29). Moreover, the authors showed high specificity rates at both 1% and 50% cut-offs, regardless of cellularity, but decreased rates in the case of <100 tumor cell samples. The authors concluded that the PD-L1 IHC of cytology cell-block specimens in NSCLC might represent a feasible alternative to histological specimens, in the case of adequate tumor cellularity, and they stressed the risk of false negative PD-L1 IHC in the case of low tumor cellularity. Therefore, cytology cell blocks represent feasible specimens to detect PD-L1 immunohistochemistry in non-small cell lung cancer, providing adequate cellularity. The analysis of low cellularity might lead to false negatives, with subsequent patient exclusion from potentially beneficial treatment [16].

Many authors evaluated the concordance between matched or unmatched cytology–histology along with other endpoints in order to analyze and compare the PD-L1 testing of feasible, minimally invasive, rapid cytology samples to specimens obtained using the recommended but more invasive biopsy or resection [17,18]. Accordingly, several and different concordance rates and k-values were provided (including separate concordance for different cut-offs and/or overall concordance), with sample size varying from 21 to 247 cases, and concordance rates from 53% [17] to almost 100% [19], due to variable rates of cellularity, the intratumoral heterogeneity of the expression of PD-L1, and more three-dimensional cell clusters in cytology samples. Shen and Li [18] demonstrated a significant heterogeneity between primary and metastatic sites and different sample types attributed to intratumoral heterogeneity but not in different tumor subtypes by evaluating the association between different specimen types and histopathological characteristics. Moreover, Gosney et al. [20] evaluated the concordance rate in a review on nine studies and a total 428 specimens, and they reported concordance rates of 88.3% and 89.7% for a tumor promotion score (TPS) cut-off of >1% and ≥50%, confirming the feasibility of cytology specimens for reliable PD-L1 evaluation. Furthermore, Sakakibara and Russel-Goldman obtained similar results by analyzing other rare clones [21,22].

## 5. Bladder Cancer and Immunity

Immunotherapy represents one of the latest and most important cancer research areas with potentially long-lasting therapeutic effects. However, not all tumors benefit from immunotherapy based on checkpoint inhibition. The potential of immunotherapy to cure most cancers, including BCa, is based on the better understanding of cancer response and resistance mechanisms. BCa is one of the most aggressive tumors, and it has been successfully treated with different immunotherapeutic approaches, such as the intravesical instillation of the Calmette-Guérin bacillus (BCG) in the early stages, and the blocking of the anti-immune action with checkpoint PD-1/PD-L1 in the late stages.

Bladder cancer (BCa) represents a model to analyze the predictors and response mechanisms of the immune system, which can be translated to other human cancers [23].

BCG therapy was the first FDA-approved immunotherapy and for over forty years it proved to be the most effective intravesical treatment in reducing the risk of BCa recurrence for high-risk diseases.

Intravesical chemo- and immune-therapy with BCG represent the universally accepted treatment for the prophylaxis of NMI-BC recurrence and progression. However, particularly for BCG, the development of these treatments is often empirical and its therapeutic mechanisms are still under investigation. They include the intact immune system, the live BCG, and the close contact of BCG with BCa cells which represent the optimal requirements for effective therapy, as well as BCG attachment and internalization, the secretion of cytokines/chemokines, and the presentation of BCG and/or cancer cell antigens to immune system cells. However, in approximately 70% of cases, BCa patients do not respond to intravesical BCG therapy [24].

In normal conditions, cancer cells would be attacked by the immune system, which would recognize them as foreign to the body. The expression of the PD-L1 protein (PD-1 protein ligand) on the surface of cancer cells and on peritumoral (tumor-associated) immune cells allows tumors to escape immune system identification, and they therefore continue to grow and proliferate [25,26]. The site of expression of checkpoint-inhibitor ligands might represent a really interesting method of contact between drug and target to explore; moreover, the direct interaction between cancer cell and checkpoint-inhibitors drugs administered intravesically might avoid the toxicity of systemic administration and might preserve and ensure direct activity.

The PD-1 inhibitors demonstrated durable antitumor activity in advanced urothelial carcinoma; moreover, the upregulation of the PD-1 pathway also was shown in NMIBC (as in the case of BCG failure) [27].

Currently, there is no widely accepted way to identify patients for which the PD-1/PD-L1 inhibitor therapy will be effective. In fact, the level of PD1 and PD-L1 expressions did not prove useful as predictors of treatment, as has been shown for other cancers [28].

Moreover, further studies failed to identify clinical characteristics representing prognostic factors for the better outcomes of immunotherapy compared to chemotherapy, or the opposite. This means that no obtained data were strong enough to contraindicate the treatment with PD-1/PD-L1 inhibitors. It should be considered not as a limitation in checkpoint inhibitor use, but as a research channel deserving exploration. Interestingly, Piao et al. showed that patients with BCG-unresponsive BCa exhibited greater PD-L1 expression than BCG-responder patients, suggesting that the PD-L1 might attenuate responses to BCG by neutralizing T-cells and show a possible biological role for PD-1/PD-L1 interactions [29].

## 6. Biomarkers for Bladder Cancer: Urinary, Blood, Tissue-Based and Others

New non-invasive diagnostic techniques have been developed in order to reach and overtake the diagnostic accuracy of cytology with a particular focus in prognosis. This issue deserves to be fully considered, especially in the case of high-risk BCa patients treated with the therapeutic gold standard (BCG). Since we cannot predict for which patients BCG therapy will fail, biomarkers with predictive value are needed.

Patient management should be tailored as much as possible, from diagnosis to treatment; this trend leads to a personalized evaluation of the biological and clinical behavior of bladder tumors, with subsequent improved oncological outcomes and optimized resource allocation.

The EORTC [30] and CUETO [31] are the most commonly used risk stratification tools, but they reported poor discrimination for both the recurrence and progression of bladder tumors; hence, there is a need for better tools incorporating more powerful predictors of cancer behavior in order to improve both the risk stratification and therapy of BCa. This missing information might be filled by integrating biomarkers, able to better reflect the biological behavior of the cell and its host.

To date, several urinary, blood, and tissue markers have been developed and tested in order to improve outcome prediction in a step towards targeted medicine; however, their suboptimal performance lead to a limited role and none of them are currently recommended by expert guidelines for daily clinical practice [32].

Urinary biomarkers are therefore used to predict short- and medium-term cancer outcomes, as well as in response to BCG.

A positive fluorescence in situ hybridization (FISH) assay performed at various time points during BCG therapy has been associated with disease persistence or recurrence. UroVysion^®^ is a multi-target assay developed for the detection of bladder tumors. Liem et al. reported a greater risk (up to 4.6 times) of developing BCa recurrence in case of a positive FISH test 3 months after the trans-urethral resection of a bladder tumor (TURBT) (alone or in combination with BCG induction) compared to a negative FISH test. The authors concluded that FISH might represent an additional tool in the decision-making process [33]. Kamat et al. reported that a positive FISH-test at both 6 and 12 weeks after the TURBT and BCG might identify the highest recurrence and progression risk rates. The results appeared promising, but need further validations [34].

Another promising biomarker of BCa is represented by the analysis of epigenetic alterations, such as DNA methylation. The methylation of DNA consists of an epigenetic variation with the affection of gene expression but without changes in DNA sequence. Several studies reported methylated loci in the BCa context, indicating its application as both a diagnostic and prognostic biomarker [35,36]. The Bladder EpiCheck (BE) is a new assay based on the profile of DNA methylation: the analysis of DNA from voiding urine allows us to detect disease-specific patterns of DNA methylation in BCa patients. In a validation study on a sample population of 222 patients, Wasserstrom et al. [37] reported an overall sensitivity of the BE of about 90%: the rates were higher in higher stages (91%, 100%, 100%, and 81% in pCIS, pT2, pT1, and pTa, respectively), and in higher grades (95% and 84% in high- and low-grade tumors, respectively). The overall specificity of BE was of about 83%, with an NPV of 97%. Conversely, compared to wash cytology (considered as the reference standard), the authors reported the greater sensitivity of the BE (90% and 38% in low- and high-grade tumors, respectively), but lower specificity (83% and 96%, respectively).

Witjes et al. [38] evaluated the BE on 353 BCa patients and reported overall rates of sensitivity and specificity of 68% (30/44) and 88% (272/309), respectively, regardless gender, age, occupational exposure, smoking habits (former or ongoing), and treatment for recent tumor recurrence. The rates of BE sensitivity were higher in higher stages (100%, 100%, and 52% for pCIS, pT1, and pTa, respectively) and grades (89% and 40% in high- and low-grade, respectively), and the NPV was of about 95% for the whole cohort of patients. In the case of HG-NMIBC, the tumor could be excluded with an NPV of 99% and could be detected with a sensitivity of 92%. The authors reported an AUC of 0.82 for all tumors (including low-grade Ta), and 0.94 when excluding low-grade recurrences.

Witjes et al. [39] analyzed the BE performance in a cohort of 357 patients in a secondary external independent analysis of the study with 13.7% of intravesical recurrences of BCa. The authors reported overall sensitivity and specificity rates of 67% and 88%, respectively, a sensitivity rate of 89% in the case of high-grade disease, NPV rates of 94% and 99% in the case of any-grade and for high-grade disease, respectively. Moreover, the authors reported that a one-point increase in the EpiScore reflected an increase of 4% in the risk of any-grade BCa, and an increase of about 8% in the high-grade NMIBC risk, by considering univariable logistic regression analyses. Furthermore, the authors associated a positive BE result independently with any-grade NMIBC (OR = 18.1, 95% CI 8.66–40.2; *p* < 0.001) and with high-grade NMIBC (OR = 78.3, 95% CI 19.2–547; *p* < 0.001) by using a multivariable logistic regression analysis. Finally, through explorative analysis, the BE performance, evaluated using the AUC, was not affected by baseline characteristics such as gender, age, pathological stage and grade, ongoing intravesical therapy, or time from last recurrence; effectively, adding the BE to the clinical variables resulted in a significant improvement in the AUC of about 16% in the prediction of any-grade BCa (BE AUC: 86%) and of 22% in the prediction of high-grade NMIBC (BE AUC: 96%).

The rates of sensitivity, NPV, and PPV of urinary cytology were compared to the BE in few studies and the diagnostic accuracy of their combination was evaluated in only one research paper [40]. Compared to the BE, the overall cytology sensitivity rates were lower, from 27% [41] up to 38% [37]; the rates were low in low-grade (0–13% across studies) and increased in high-grade NMIBC (50–67% across studies) while remaining lower than the BE. Notably, the NPV rate was higher in the BE compared to cytology in all studies. Hence, thanks to the high sensitivity and NPV, the BE might be considered as a useful tool for the improvement of the sensitivity of cytology.

Pierconti et al. [42] applied the BE evaluation in a cohort of 374 cases of high-grade NMIBC (268 patients with pT1 and 106 CIS patients) according to different cytological reports (negative for high-grade urothelial carcinoma, NHGUC; atypical urothelial cells, AUC; suspicious for high-grade urothelial carcinoma, SHGUC; high-grade urothelial carcinoma, HGUC; low-grade urothelial neoplasia, LGUC; unsatisfactory/non-diagnostic), and the authors reported how differences between cytological categories might even be based on molecular disparities. The authors showed how the EpiScore value increased from NHGUC to HGUC: an EpiScore value lower than 60 correlated with NHGUC (OR = 3.9, 95%CI 1.9–8.1, *p* = 0.0003) and a value higher than 60 with SHGUC (OR = 3.8, 95%CI 1.6–8.9; *p* = 0.0031) and HGUC (OR = 3.9, 95%CI 1.6–9.5; *p* = 0.0027). Moreover, after 1 year, the authors reported rates of sensitivity, specificity, and NPV for the BE of 100%, 89.9%, and 100%, respectively, in the NHGUC group, and 98%, 100%, and 86%, respectively, in the HGUC group. The combination of the BE and cytology could increase the sensitivity of each test, especially in the follow-up of high-grade NMIBC, with a potential subsequent reduction in the number and frequency of cystoscopies, especially in the case of low-risk disease. Nevertheless, the BE analysis needs dedicated and equipped laboratory facilities and an experienced pathologist to perform a real time PCR.

Blood-based biomarkers measuring systemic inflammatory response such as the neutrophil-to-lymphocyte ratio (NLR) and C-reactive protein (CRP) were also evaluated as predictors of cancer outcomes in NMIBC. Based on the neutrophil-to-lymphocyte ratio (NLR) and the C-reactive protein (CRP) as markers of systemic inflammatory response, associated with the prognosis of multiple malignancies, Mbeutcha ed al. aimed to correlate them to the oncologic outcomes of NMIBC. The authors retrospectively reviewed the medical records of 1117 NMIBC patients who underwent TURBT with a median follow-up at 64 months. They found a high NLR (≥2.5) in 360 patients (32.2%) and a high CRP (≥5 mg/L) in 145 patients (13.0%). The authors showed an association between a high NLR and both disease recurrence (subhazard ratio [SHR] = 1.27, *p* value = 0.013) and progression (SHR = 1.72, *p* value = 0.007), and between a high CRP and disease progression (SHR = 1.72, *p* value = 0.031) through multivariable analyses. Moreover, the multivariable model predicting disease progression lead to a relevant change in discrimination (+2.0%), with the addition of the NLR to the CRP. A high NLR and a high CRP were both associated with disease progression (SHR = 2.80, *p* value= 0.026 and SHR = 3.51, *p* value = 0.021, respectively), and the NLR to disease recurrence (SHR = 1.46, *p* value = 0.046), in a subgroup analysis of 300 BCG-treated patients. The authors even reported an increase in the discrimination of the model predicting progression after BCG therapy with the inclusion of both markers (+2.4%) in the evaluation. In NMIBC patients, markers of systemic inflammation response were therefore associated with the recurrence and progression of the disease. The inclusion of such markers in prognostic models might enhance their accuracy and might lead to an increase in model discrimination. These biomarkers are surely of interest, and they might be able to select patients who are more likely to benefit from systemic immunotherapy such as checkpoint inhibitors [43].

Based on the lack of accurate systems of classification able to predict recurrence or progression risks, Amantini et al. evaluated circulating tumor cells (CTCs) isolated by ScreenCell devices from MIBC and NMIBC patients in order to build a sort of prognostic gene signature. The authors selected a 15-gene panel modulated in BCa compared to controls, evaluated their expression in CTCs, and recognized EGFR, TRPM4, TWIST1, and ZEB1 as prognostic biomarkers via univariate and multivariate Cox regression analyses. Moreover, the 4-gene signature allowed them to significantly group patients into high and low risk in RFS terms (HR = 2.704, 95%CI = 1.010–7.313, Log-rank *p* value < 0.050) with subsequent improvement in the choice of the best treatment for BCa [44].

Regarding tissue-based biomarkers, different genes and proteins were reported to be related to different pathways involved in both bladder carcinogenesis and its clinical behavior. Despite this, several tissue biomarkers were evaluated via a systematic multiphase approach (cell cycle markers such as Ki-67, FGFR3, cadherin, and immune- and inflammation-related biomarkers) and they proved that they could predict BCa outcomes, but their prognostic value remained not optimal. Therefore, only a few were considered in subsequent prospective validation study phases [45,46,47,48,49,50]. In a prospective study, Van Kessel et al. studied a panel of tissue-based biomarkers: the authors compared the accuracy performance of the panel to the clinical–pathological characteristics used for risk stratification [51]. The authors analyzed GATA2, TBX2, TBX3, and ZIC4 methylation and FGFR3, TERT, PIK3CA, and RAS mutation status in fresh-frozen tumor samples from 1239 patients with primary or recurrent non-muscle-invasive BCa. Overall, the authors reported a significant association among disease progression and wild-type FGFR3 and the methylation of GATA2 and TBX3. The addition of these selected markers increased the accuracy of the EORTC risk stratification model, allowing them to identify selected patients with a very high risk for disease progression who could benefit from tailored therapy such as intensified combination systemic therapy or early radical cystectomy.

In BCa, a single biomarker is unlikely to meet the unmet needs due to several mutations and the intratumoral heterogeneity of physicians. Differently, biomarker panels could represent useful tools in the stratification of patient risk and treatment selection by integrating many pathways involved in diagnosis, staging, prognosis, and/or prediction [32].

The BCa emits several volatile organic compounds (VOCs) in the urine headspace and they can be analyzed using an electronic nose. Bassi et al. studied the diagnostic performance of the electronic nose (32 volatile gas analyzer sensors) applied in BCa through a pilot, prospective, single-center, controlled, non-randomized, phase II study on 102 patients with proven BCa and 96 controls. The authors evaluated accuracy, sensitivity, specificity, and variability by using a non-parametric combination method, permutation tests, and discriminant analysis classification. The authors showed statistically significant differences between BCa patients and controls by using 28 of the 32 sensors (87.5%) and rates of overall discriminatory power, sensitivity, and specificity of 78.8%, 74.1%, and 76%, respectively. Moreover, the authors reported rates of misclassification of 13.5% (13/96) in the control group (as a false positive) and 28.4% (29/102) in BCa patients (as a false negative). Through the selection of the most efficient sensors, the rates of sensitivity and specificity increased up to 91.1% (72.5–100) and 89.1% (81–95.8), respectively, but none of the tumor characteristics were proven as independent predictors of device responsiveness. The authors highlighted the advantages of the electronic nose as a potentially reliable, quick, accurate, and cost-effective tool for the non-invasive diagnosis of BCa [52].

Multiple molecular subtypes of BCa were studied via DNA/RNA-based classification, but the evidence is scarce regarding the protein level. Stroggilos et al. evaluated an NMIBC stratification into biologically meaningful groups based on the proteome. The authors processed it for high-resolution proteomic analysis using liquid chromatography–tandem mass spectrometry (LC-MS/MS) tissue specimens from 98 patients with NMIBC and 19 with MIBC at primary diagnosis. The proteomics output underwent unsupervised consensus clustering, principal component analysis (PCA), and the investigation of subtype-specific features, pathways, and gene sets. The authors stratified patients with NMIBC into three proteomic subtypes (NPS), differing in size and clinicopathologic and molecular backgrounds: NPS1 (mostly high-stage/grade/risk samples) (17/98 patients) with overexpressed proteins as in an immune/inflammatory phenotype involved in DNA damage response, cell proliferation, and unfolded protein response; NPS2 (mixed-stage/grade/risk composition) with an infiltrated/mesenchymal profile; and NPS3 rich in luminal/differentiation markers (in line with pathological composition, with mostly low-stage/grade/risk samples). In the PCA, the authors reported a close proximity of NPS1 and, conversely, remoteness of NPS3 to the MIBC proteome. Moreover, the authors reported that proteins distinguishing NPS1 and 3 consistently differed at the mRNA levels between high- and low-risk subtypes [53].

Urine metabolomics proved to be a feasible approach to detect potential biomarkers for cancer diagnosis. Wang et al. used an ultra-performance liquid chromatography coupled to mass spectrometry (UPLC-MS) method in the evaluation of the urinary metabolites from 29 patients with BCa and 15 healthy controls. The authors extracted the differential metabolites and analyzed them using univariate and multivariate analysis methods: 19 metabolites were extracted as differently expressed biomarkers in the two groups, and then mainly related to the pathways of phenylacetate metabolism, propanoate metabolism, fatty acid metabolism, pyruvate metabolism, arginine and proline metabolism, glycine and serine metabolism, and bile acid biosynthesis. Moreover, a subgroup was created of 11 metabolites of those 19 revealed as potential biomarkers for BCa diagnosis by using a logistic regression model. The authors reported rates of area under the curve (AUC) value and the sensitivity and specificity of the receiving operator characteristic (ROC) curve of 0.983, 95.3%, and 100%, respectively, supposing a very high discrimination power for BCa patients from healthy controls. It was the first time where the potential diagnostic markers of BCa via metabolomics was revealed, and this provided a new site for exploring biomarkers in future research [54].

## 7. Rationale for Immune Checkpoint Inhibitors in Bladder Cancer

T-cell-mediated immunity consists of sequential phases: the clonal selection of APC and the activation, proliferation, transition, and implementation of direct effector function. These phases represent a balance between inhibitory and stimulatory signals [55]. In a non-tumor environment, immune checkpoint proteins are responsible for controlling the immune system and prevent autoimmunity by following inhibitory pathways that physiologically counterbalance the co-stimulatory pathways to appropriately adjust immune responses [56].

Cancer cells can evade antitumor immunity by adopting several escape strategies such as diminishing MHC-I expression, and consequently, CD8+ T-cell activity; defective antigen processing and presentation, and consequently, a reduced recognition by T-cells; and increasing the expression of co-inhibitory (i.e., immune checkpoint) molecules [57]. Most cancers use the immune checkpoints to evade immune system attack by blocking the effector T-cell functions; hence, antitumor immunity might be enhanced and/or recovered by antibodies that inhibit the receptor–ligand interaction and deactivate the immune checkpoints [58]. Currently, PD-1, PD-L1, and CTLA-4 represent the most investigated and clinically related immune checkpoint molecules.

The wide mutational spectrum and the heterogeneity of urothelial cancer surely represent advantages in adopting efficient immunotherapies: the mutations induce several neo-antigens, recognized as ‘non-self’ by the circulating T-cells, thereby inducing the immune response [59]. A high mutational burden was observed in urothelial cancer, melanoma, and non-small cell lung cancer [60], and the high tumor mutational burden was identified to be proportional to the clinical efficacy of the PD1-L1 blockade, especially in melanoma and non-small cell lung cancer [60,61]. Consequently, several monoclonal antibodies were developed and clinically applied for the management of urothelial cancer, including bladder and upper urinary tract cancers [62,63,64,65].

## 8. Role of PD-1/PD-L1 in Bladder Cancer

In the study of Kawahara et al., PD-1 and PD-L1 were more and highly expressed in high-grade BCa compared to low-grade cases; the authors highlighted the potential correlation between PD-1/PD-L1 expression and the pathological BCa grade as an effective biomarker. Moreover, PD-L1 might even represent a predictor of stage progression in bladder tumors [66].

Pierconti et al. analyzed the expression of PD-L1 in primary CIS in BCG-failure and BCG-responder BCa patients. In tumor cells and in immune cell compartments, the expression of PD-L1 was more often detected in BCG-failure patients. In their study, only the expression of PD-L1 22C3 in tumor cells correlated with tumor recurrence. The authors concluded that PD-L1 22C3 expression might identify BCG-non-responder CIS cases, supporting the hypothesis that high intratumoral levels of PD-L1 22C3 might explain the BCG failure [67].

Kates et al. evaluated the characterization of the expression of immunity cells among patients with BCG-treated NMIBC: the adaptive immune resistance represented a mechanism of BCG failure, with baseline tumor PD-L1 expression predicting an unfavorable BCG response; if validated, baseline tumor PD-L1 expression might be used to guide therapeutic decision. [68]

Fukumoto et al. analyzed the variation of PD-1 expression before and after treatment with BCG and its role in prognostic terms in NMIBC. The BCG therapy itself lead to PD-1 expression, and this might represent a valuable prognosticator of worse clinical outcomes in NMIBCa BCG-treated patients [69].

Hashizume et al. reported how the PD-L1 expression was enhanced on tumor tissue after BCG treatment in BCG-resistant NMI-BCa patients. In this cohort of patients, the authors speculated how immunotherapy with anti-PD-1/PD-L1 antibodies could be feasible when combined with BCG or as secondary treatment at recurrence after BCG [70].

In a study on 407 BCa patients, Xu et al. demonstrated the prognostic value of CD274 (PD-L1 promoter gene) methylation in BCa patients. PD-L1 methylation was revealed to be an independent predictor for OS (*p* = 0.037). Moreover, PD-L1 methylation might be considered a prognostic biomarker for immunotherapy response. However, PD-L1 methylation and PD-L1 mRNA expression were not statistically associated with chemotherapy response [71].

These authors evaluated the expression of checkpoint inhibitors exclusively on histological specimens.

## 9. New Immunotherapy: Intravenous and Intravesically Administered

BCa is a highly immunogenic cancer [72,73] and one of its treatments is based on cancer immunotherapies stimulating the body’s immune system (such as BCG) [68].

New classes of immune checkpoint inhibitors have been developed in the past decade (Pembrolizumab, Atezolizumab, Nivolumab, Avelumab, Durvalumab, Ipilimuumab, Tremelimab, etc. [29]).

The NCT02324582 trial evaluated the efficacy of intravenous immune checkpoint inhibitors in combination with BCG in patients with NMIBC [74]. Moreover, NCT02451423 [75] and NCT02450331 [76] clinical trials evaluated the intravenous (neo-adjuvant and adjuvant) immune checkpoint therapy in patients treated with cystectomy.

For several decades, BCG has been used to reduce the recurrence risk of high-risk NMIBC: Intravesical BCG was the first FDA-approved immunotherapy and represented the most effective treatment. However, up to 70% of BCa patients might not respond to BCG [77]. Interestingly, BCG-non-responder patients exhibit higher PD-L1 expression than BCG-responder patients. This suggests how PD-L1 could attenuate responses to therapy with BCG by neutralizing T-cells and, conversely, it could possibly play a biological role in PD-1/PD-L1 interactions [65].

Several studies on (new) immunotherapy in BCa patients based on intravenous administration as a second-line approach and/or in advanced stages, while early stages such as non-muscle-invasive BCa have not been extensively studied yet. Currently, despite the intense testing of checkpoint inhibitors, few studies focused on an intravesical way of administration of these new immunotherapies in patients with refractory NMI-BCa after standard treatment and therefore candidates for radical cystectomy. As phase I or II studies, these research papers primarily have safety and tolerability as endpoints.

The FDA approved Durvalumab and Pembrolizumab for the treatment of locally advanced or metastatic urothelial carcinoma; therefore, they are still considered as investigational agents in the setting of intravesical administration.

Hence, all these trials include correlative studies of pharmacokinetics, humoral/cellular responses, and potential biomarkers (including PD-1/PD-L1) in blood and tumor tissue before and after intravesical checkpoint inhibitor instillation in order to fully explore this new approach.

In ongoing trials, NCT02808143 [78], NCT03167151 [79], NCT03759496 [80], Pembrolizumab (anti PD-1), and Durvalumab (anti PD-1) are administered intravesically, alone, or in sequential combination with BCG. Interestingly, in ACTRN12620000063910 trials [81], Durvalumab is administered with a sub-urothelial injection rather than as a passive intravesical solution, opening up new and remarkable drug delivery opportunities. These administration modalities surely sum up the experience in the field, especially in relation to sequential or device-assisted treatments [82].

Another issue explored is to determine the maximum tolerated dose (MTD), previously one of the most studied aspects for standard intravesical therapies and certainly one of the benchmarks of the future development of these intravesically administered drugs [78,80].

The intravesical administration allows for several advantages: water solubility, higher concentration, easy administration, selective activity, free drug in a fully active form, and mostly local side effects (with increased tolerability). The intravesical administration of the standard therapy is generally well tolerated [83], but the toxicity analysis will certainly represent a further element of comparison between the two approaches (standard vs. new immunotherapy). Currently, they represent further promising and challenging issues to fully explore.

Furthermore, since these studies are currently focused only on patients who did not respond to standard therapy, intravesical checkpoint inhibitors might represent a valid alternative therapy in case of intolerance to standard agents, thus also overcoming the BCG shortage issue.

It was not established whether immunity activated via intravesical and systemic administration is the same; by the same token, we can only speculate that non-systemic administration might be better in terms of immune-related adverse events.

In these perspectives, the use of anti-PD1/PD-L1 drugs is encouraged in order to more comprehensively explore the role of the immune system in BC, with the opportunity of more targeted treatments (the right therapy for each patient). The checkpoint inhibitors represent an important development in the treatment of urothelial cancer. On the basis of the clear benefits shown through intravenous systemic administration, treatment with checkpoint inhibitors might also be taken into consideration for intravesical administration. Non-muscle-invasive bladder cancer might greatly benefit from both the past experience and the new evidence in immunotherapy, and this research deserves to be developed.

## 10. Combination of PD-L1 Expression with New Treatments

Among emerging intravesical treatments, gene therapy with nadofaragene firadenovec has provided promising results (with a 35% rate of high-grade recurrence-free status at 12 months) [14], and an ongoing phase III trial is an FDA registration trial [Clinicaltrials.gov. Bethesda: US National Library of Medicine; c2016 [updated 24 May 2018]. https://clinicaltrials.gov/ct2/show/NCT02773849] (accessed on 1 August 2023).

With the view of combining and translationally linking new treatment alternatives to standard therapies (such as BCG or Pembrolizumab), Mitra et al. [84] evaluated the PD1 and PD-L1 status in TURBT specimens from patients treated with nadofaragene firadenovec. The authors evaluated the role for combining nadofaragene with anti-PD1 therapy in BCG-unresponsive NMIBC. There are few efficient bladder-preserving therapies for BCG-unresponsive NMIBC, and Pembrolizumab (checkpoint-inhibitor-targeting PD1 receptor) is approved in this setting. Nadofaragene firadenovec is an intravesical adenoviral vector-based therapy (it delivers IFNA2 to urothelial cells), with proven durable responses in a multicenter phase III trial on BCG-unresponsive NMIBC patients (# NCT02773849). The authors evaluated TURBT specimens of 85 patients treated on a single-arm phase III trial with nadofaragene with the following schedule: once every 3 months for up to 4 doses. PD-1 (positive, >0% cells) and PD-L1 (positive, ≥1% cells) quantifications were performed both for urothelial and infiltrating lymphocyte compartments. The authors treated orthotopic tumors in C57Bl/6 mice with intravesical adenoviral-IFN versus control, and they assessed the PD-1/PD-L1 status. Urothelial PD1 and PD-L1 status assessed on study-entry (Sin) and study-exit (Sout) TURBT specimens were not associated with treatment response (all, *p* ≥ 0.26). Sin lymphocyte PD-1+ was detected in 52% and 57% of responders (R) and of non-responders (NR) (*p* = 0.72), respectively, while Sin lymphocyte PD-L1+ in 77% and in 92% of R and NR (*p* = 0.15), respectively. Moreover, in 62% of NR, the authors reported Sout lymphocyte PD-1+ compared to 26% R (*p* = 0.002), and Sout lymphocyte PD-L1+ was also revealed to be higher in NR compared to R (78% versus 50%, *p* = 0.024). The authors reported statistically significant differences in Sout lymphocyte positivity status between R and NR as early as 3 months after therapy for PD-1 (*p* = 0.040) and PD-L1 (*p* = 0.029). The PD-1 overexpression was reported in co-expression assays on orthotopic bladder tumors on T lymphocytes in control mice compared to those that responded to intravesical adenoviral-IFN. In conclusions, nadofaragene-non-responder patients presented relative PD1/PD-L1 overexpression in tumor-infiltrating lymphocytes, despite the fact that this was not evident at baseline. In vivo, these findings were replicated. The authors highlighted the role of immune checkpoint inhibitors in BCG-unresponsive NMIBC who may have a partial or short-term response to nadofaragene [85].

## 11. Urinary PD-L1 Detection Methods

In their study, Tosev et al. stored all urine samples at room temperature; they used whole urine without centrifugation (tumor and control samples were stored for the same length of time). The authors analyzed 100 μL of urine with the ELISA, performing experiments in duplicates. A standard curve for each experiment was created with assay performance, and in the data analysis only experiments with a linear curve were included. In order to measure PD-L1 in the urine samples, a Quantikine ELISA for Human/Cynomolgus Monkey PD-L1/B7-H1 Immunoassay from R&D Systems was used (Catalog Number DB7H10) [86].

In their study, Vikerfors et al. centrifuged urine samples at 2000 rpm for 10 min, and stored them at −80 °C before analysis. The authors retrospectively analyzed all the samples, and cases and control samples were simultaneously processed: soluble PD-L1 (sPD-L1) in urine measured with a commercially available ELISA for PD-L1 was included in each ELISA with control samples with known PD-L1 concentrations (low, medium, and high). The authors reported a minimum detectable sPD-L1 level of about 4.52 pg/mL (range 25.0–1600), with range of coefficient of variation (CV) of about 0–10%; in a 4-Parameter Logistic non-linear regression analysis on values of absorbance vs. concentrations, the R2 values above 0.9 were considered as acceptable [87].

## 12. Comparison to the Literature Evidence

Only a few studies focused on the analysis of the expression of PD-L1 in the urine of BCa patients (Table 1). To our knowledge, there is no previously published evidence on the evaluation by percentage expression of PD-L1 in urine.

In a proof-of-concept study, Tosev et al. showed how the PD-L1 expression can be measured in the urine samples of NMIBC and MIBC patients, and that urine PD-L1 levels were significantly higher in NMIBC and MIBC patients when compared to healthy controls. The median urine levels of PD-L1 were higher in both newly diagnosed (11.28 pg/mL and 71.73 pg/mL in NMBIC and MIBC, IQR: 0–21 and 27–123, respectively) and recurrent BCa patients (7.9 pg/mL and 4.1 pg/mL in NMBIC and MIBC, IQR: 0–20 and 0–12, respectively) compared to the control group (0 pg/mL, IQR: 0–3) (*p* < 0.05 to *p* < 0.01). In the post-TURBT group, the authors reported no statistically significant differences via a direct comparison of the PD-L1 urine level between patients with a documented negative cystoscopy (N = 13) and the other post-TURB patients (N = 50) (median 4.45 pg/mL, IQR: 1–14 vs. median 5.93 pg/mL, IQR: 0–20; *p* = 0.99). The authors showed the diagnostic profile of urinary PD-L1 using a ROC curve analysis, with the optimal cut-off identified using the Youden index. The threshold PD-L1 concentration was calculated as 10.05 pg/mL and 2.95 pg/mL for comparison between controls and newly diagnosed pre-TURB BCa patients and recurrent post-TURB BCa patients, respectively: the AUC was highest (0.78) when used for the detection of newly diagnosed BCa patients, with a sensitivity and specificity of 0.65 and 0.95 and PPV and NPV of 0.87 and 0.84. Furthermore, these values appeared in line with published results for urinary protein biomarker tests approved by the U.S. FDA. Despite the fact that no single protein biomarker has yet achieved the optimal diagnostic accuracy, the authors reported that PD-L1 could potentially be considered as a valuable biomarker in addition to the multiparametric panel for the monitoring and potential detecting of bladder tumors [86].

Vikefors et al. evaluated the levels of sPD-L1 in the serum and urine samples of 132 patients with BCa compared to 51 controls using ELISA. The authors found sPD-L1 in 99.5% and 34.4% of serum and urine samples, respectively, with a median concentration of urinary sPD-L1 of 74.2 pg/mL (range 57.5–669.2). The authors did not report a statistical correlation among serum and urinary sPD-L1 levels (R = 0.167, *p* value = 0.22), nor between urinary sPD-L1 and age, smoking status, or BMI. The authors reported higher urinary sPD-L1 levels in BCa patients compared to controls, but no difference in serum sPD-L1 levels (*p* values of 0.038 and 0.61, respectively). Moreover, urinary sPD-L1 seemed to be more frequently identified in patients with BCa than controls (*p* value = 0.07). The authors showed no statistically significant associations among urinary sPD-L1 levels and pT-stage and grade (low vs. high) (*p* values of 0.09 and 0.09, respectively). However, patients with metastatic disease at initial diagnosis or during follow-up had higher of urinary sPD-L1 levels compared to patients without metastases (*p* value = 0.05). No association between sPD-L1 levels in urine and all-cause mortality was reported (*p* = 0.09). In conclusion, only serum (but not urinary) sPD-L1 might be considered as a biomarker with a potential prognostic role in BCa cases [87].

## 13. Perspectives

Optimization and standardization of the technique (from specimen collection to preparation—including staining—and interpretation of results).Enhancement of urinary cytology diagnostic accuracy (is it always able to detect urothelial cancer?).Identification of the standard Combined Positive Score (CPS) in cytological specimens of BCa.Development of new, non-invasive tools for bladder cancer management with prognostic characteristics.Evaluation of therapy response by analyzing pre- and post-treatment PD-L1 expression.Identification of patients who might benefit from systemic checkpoint inhibitor therapy.Identification of patients who might benefit from intravesical checkpoint inhibitor therapy.

## 14. Conclusions

PD-L1 expression might be detected even in urine samples in BCa patients, in addition to the examination of a histological sample. The technique is being standardized and optimized. We reported that BCa patients had higher urinary PD-L1 levels than controls by considering BCa tumors expressing PD-L1 in the tissue specimen. The expression of PD-L1 on urinary BCa cells might represent both a diagnostic and a prognostic tool, with the perspective that the PD-L1 expression on exfoliate urinary cells might reveal and anticipate eventual BCa recurrence or progression.

The quantification of PD-L1 expression in urinary samples through the use of consolidated methods for other analyses such as ELISA is able to give a numerical value to the expression, and as such continuous data, resulting in many findings in diagnostic, prognostic, and follow-up terms. In addition to measurement as a concentration, PD-L1 expression could also be expressed in percentage terms, as is the case for other tumors. In both cases, identifying absolute numerical and percentage values can pave the way for identifying cut-offs, as well as for the Epicheck test.

Further prospective and longitudinal studies are needed to assess the expression of PD-L1 as a biomarker for the monitoring of BCa patients.

## Figures and Tables

**Figure 1 diagnostics-13-03119-f001:**
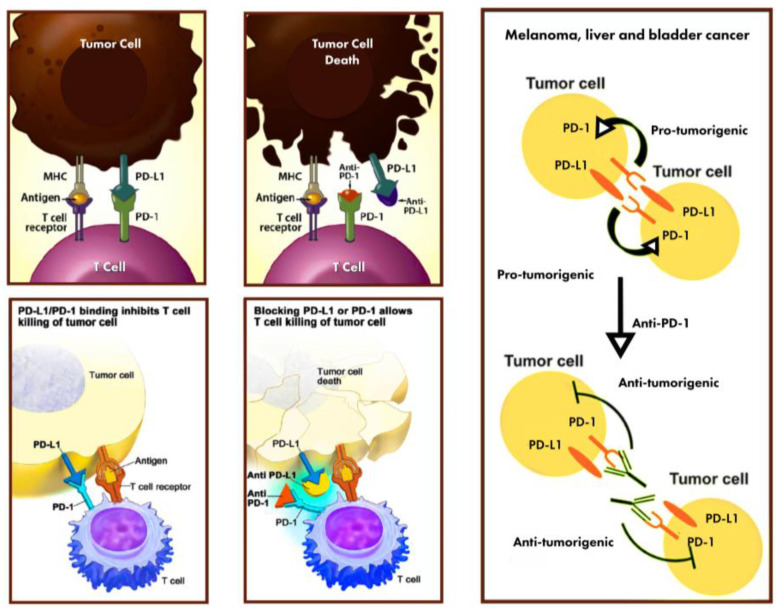
The role of PD-L1 expression in BCa tumors.

**Table 1 diagnostics-13-03119-t001:** Comparison of the available literature.

	Method	No. of Patients	Median PD-L1 Expression in Urine	*p* Value
Vikerfors et al. [87]	A commercially available ELISA for PD-L1: R&D System Inc., Minneapolis, MN, USA; Catalogue no. DB7H10 for BCa Patients, no. QC226 for controls	Cases: 45	75.7 pg/mL (60.6–669.2)	0.038
		Controls: 11	70.0 pg/mL (10.7–57.5)	
Tosev et al. [86]	A Quantikine ELISA for Huma/Cnomolgus Monkey PD-L1/B7-H1 Immunoassay from R&D Systems	NMIBC: 47 MIBC 36	9.55 pg/mL (IQR 0–20) 5.93 pg/mL (IQR 0–17)	<0.001
		Control: 39	0 pg/mL (IQR 0–3)	

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
