# Peer review of "The Role of Checkpoint Inhibitor Expression Directly on Exfoliated Cells from Bladder Cancer: A Narrative Review"

_diagnostics, 2023, doi:10.3390/diagnostics13193119_

Round 1

Reviewer 1 Report

Bladder cancer (BCa) is a prevalent form of cancer impacting the urinary bladder. Detecting and managing BCa at an early stage is crucial for successful treatment and favorable patient results. One such biomarker is programmed death-ligand 1 (PD-L1), found on the surface of cancer cells, and it plays a vital role in evading the immune system. Research has indicated that PD-L1 expression is higher in BCa tumors compared to healthy bladder tissue. Furthermore, PD-L1 expression may be detectable in urine samples from BCa patients, supplementing the examination of histological samples. Efforts are underway to standardize and optimize this technique. This study revealed that BCa patients exhibit elevated levels of urinary PD-L1 compared to controls, considering PD-L1 expression in tissue specimens from BCa tumors. The presence of PD-L1 on urinary BCa cells could serve as both a diagnostic and prognostic tool, potentially anticipating BCa recurrence or progression. Further prospective, long-term studies are required to assess the potential of PD-L1 as a biomarker for monitoring BCa patients. Utilizing PD-L1 as a biomarker for BCa detection and monitoring holds the promise of significantly enhancing patient outcomes by enabling earlier detection and more efficient disease management. From a reviewer's point of view, the review is well-structured and well-organized. it can be accepted for publication after major revision includes the following:

1. Add a figure in the intro section describing the role of PD-L1 expression in BCa tumors.

2. Expand the conclusion section with more details about the new findings that came out of this review study. 

3. please update the references section with more recent artciles.

No major issue

Author Response

Reviewer 1

Thank you for your valuable revision.

We followed all the suggestions as following:

1.Add a figure in the intro section describing the role of PD-L1 expression in BCa tumors.

We added a figure as requested

2.Expand the conclusion section with more details about the new findings that came out of this review study. 

We expanded the conclusion section with more details about the new findings that came out of this review study. 

“The quantification of PD-L1 expression in urinary samples through the use of consolidated methods for other analyzes such as ELISA, is able to give a numerical value to the expression, and as such a continuous data, resulting in many findings in diagnostic, prognostic and follow-up terms. In addition to measurement as a concentration, PD-L1 expression could also be expressed in percentage terms, as is the case for other tumors. In both cases, identifying absolute numerical and percentage values can pave the way for identifying cut-offs, as well as for the epicheck test.”

3.please update the references section with more recent articles.

We updated some references with more recent ones; we decided to maintain some other references even not updated but considered as relevant in the research process.

1.Bakkar, Rania, et al. "Impact of the Paris system for reporting urine cytopathology on predictive values of the equivocal diagnostic categories and interobserver agreement." Cytojournal 16 (2019).

5.Wojcik , E.M., et al. The Paris System for Reporting Urinary Cytology. 2022, Springer.

8.Nikas IP, Seide S, Proctor T, Kleinaki Z, Kleinaki M, Reynolds JP. The Paris System for Reporting Urinary Cytology: A Meta-Analysis. J Pers Med. 2022 Jan 27;12(2):170. doi: 10.3390/jpm12020170. PMID: 35207658; PMCID: PMC8874476.

Reviewer 2 Report

1. Regarding the research progress of PD-L1 expression and as a marker in different pathological types of bladder tumors, please supplement this section.

2.Line 623 and 625 are repeated.

Author Response

Reviewer 2

Thank you for your valuable revision.

We followed all the suggestions as following:

1.Regarding the research progress of PD-L1 expression and as a marker in different pathological types of bladder tumors, please supplement this section.

Regarding the diagnostic potential value of PD-L1, we focused on this specific issue in section n°8; we reported the different and variegate data reporting experiences, we highlighted the ongoing sparse results from the different studies; in this light, we tried to be open-mind to different data and we reported them, even though some reports might appear confounding and in contrast with other ones; this means the necessity of further and based on different methods studies in order to make enrich the knowledge about the PD-L1 expression.

2.Line 623 and 625 are repeated.

We deleted as suggested.

Reviewer 3 Report

This article introduces checkpoint inhibitors expression directly on exported cells from bladder cancer, which has good clinical and basic research significance. The description of this article is clear and detailed, and it is recommended to pub

Author Response

Thank you for your valuable revision.

Reviewer 4 Report

nicely written narrative review on the concept of using CKI expression on urinary system liquid samples - urine or wash specimens.

unfortunately only a small part of it is dedicated to this topic, stated in the article - paragraphs 11 and 12.

in this reviewer`s opinion this nicely written manuscript should be reformatted, it main subject being the role of PD as a whole

Author Response

Thank you for your valuable revision.

We followed all the suggestions as following:

nicely written narrative review on the concept of using CKI expression on urinary system liquid samples - urine or wash specimens.

unfortunately only a small part of it is dedicated to this topic, stated in the article - paragraphs 11 and 12.

in this reviewer`s opinion this nicely written manuscript should be reformatted, it main subject being the role of PD as a whole.

Unfortunately, the topic is still debated and under consideration from different and sparse points of view, also in terms of different researches with subsequent different analysis methods. One of the aim of this paper was to highlight the potential of this type of research, in which for the first times the expression of PD-L1 on exfoliated urinary cells was considered to be studied; the sparse data, in our opinion, don’t mean as limitation, but we considered them as real challenging prospectives; one of the most important part of this paper was a sort of narrative review and demonstration on ongoing different analysis methods of urinary specimens, and how long was the process for each of them; therefore, they represents the basis of the proposed analysis method which in our opinion deserve to be fully and comprehensively explored. The exclusive focus on studies on expression of PD-L1

Round 2

Reviewer 1 Report

The paper can be accepted in its current status.

No major issues

Reviewer 4 Report

authors haven`t made any significant changes